# Development of an rpS6-Based Ex Vivo Assay for the Analysis of Neuronal Activity in Mouse and Human Olfactory Systems

**DOI:** 10.3390/ijms252313173

**Published:** 2024-12-07

**Authors:** Emma Broillet-Olivier, Yaëlle Wenger, Noah Gilliand, Hugues Cadas, Sara Sabatasso, Marie-Christine Broillet, Julien Brechbühl

**Affiliations:** 1Faculty of Medicine Hradec Králové, Charles University, 500 00 Hradec Králové, Czech Republic; stoope@lfhk.cuni.cz; 2Department of Biomedical Sciences, Faculty of Biology and Medicine, University of Lausanne, Bugnon 27, CH-1011 Lausanne, Switzerland; yaelle.wenger@unil.ch (Y.W.); noah.gilliand@unil.ch (N.G.); 3Faculty of Biology and Medicine, University of Lausanne, Bugnon 9, CH-1005 Lausanne, Switzerland; hugues.cadas@unil.ch (H.C.); sara.sabatasso@unil.ch (S.S.); 4Faculty Unit of Anatomy and Morphology, University Center of Legal Medicine Lausanne-Geneva, Lausanne University Hospital and University of Lausanne, Vulliette 4, CH-1000 Lausanne, Switzerland

**Keywords:** olfaction, olfactory subsystems, Grueneberg ganglion, neuronal activity, rpS6, environmental factors, 3Rs, mouse, human

## Abstract

Olfactory sensitivity to odorant molecules is a complex biological function influenced by both endogenous factors, such as genetic background and physiological state, and exogenous factors, such as environmental conditions. In animals, this vital ability is mediated by olfactory sensory neurons (OSNs), which are distributed across several specialized olfactory subsystems depending on the species. Using the phosphorylation of the ribosomal protein S6 (rpS6) in OSNs following sensory stimulation, we developed an ex vivo assay allowing the simultaneous conditioning and odorant stimulation of different mouse olfactory subsystems, including the main olfactory epithelium, the vomeronasal organ, and the Grueneberg ganglion. This approach enabled us to observe odorant-induced neuronal activity within the different olfactory subsystems and to demonstrate the impact of environmental conditioning, such as temperature variations, on olfactory sensitivity, specifically in the Grueneberg ganglion. We further applied our rpS6-based assay to the human olfactory system and demonstrated its feasibility. Our findings show that analyzing rpS6 signal intensity is a robust and highly reproducible indicator of neuronal activity across various olfactory systems, while avoiding stress and some experimental limitations associated with in vivo exposure. The potential extension of this assay to other conditioning paradigms and olfactory systems, as well as its application to other animal species, including human olfactory diagnostics, is also discussed.

## 1. Introduction

In animals, the ability to detect and discriminate thousands of odorant molecules in the environment is mediated by olfactory sensory neurons (OSNs). These neurons, depending on the species, are distributed across several olfactory subsystems, each with its own sensory specializations, morphological characteristics, and physiological regulations [1,2]. In mice (*Mus musculus*), the olfactory system includes the main olfactory epithelium (MOE), primarily involved in detecting volatile substances, such as food-derived molecules that are innately hedonic or repulsive [3,4]. These neurons, in direct contact with the nasal cavity, are distributed along the vault (dorsal part) and the nasal septum (septal organ of Masera), as well as on lamellar cartilaginous structures, commonly called turbinates, which help regulate the airflow carrying olfactory information. The Jacobson’s organ, or vomeronasal organ (VNO), is primarily involved in detecting less volatile or non-volatile molecules, such as pheromones, which are crucial for intra-species communication [4,5]. This organ, encased by a cartilaginous tube, is connected to the outside through its own access, which forms the lumen, requiring active physical contact between the animal and the pheromone-emitting source [6,7,8]. The Grueneberg ganglion (GG), on the other hand, is mainly involved in detecting molecules associated with danger, such as alarm pheromones emitted by distressed conspecifics or kairomones involuntarily secreted by predators [9,10,11]. This olfactory subsystem, protected from the outside by a keratinized epithelium permeable to water-soluble volatile molecules, is localized at the entrance of the nostrils at the rhinarium (muzzle) of the mouse. This structural feature makes it dependent on the surrounding temperature, directly affecting its sensory sensitivity [12,13,14].

Currently, the functional investigation techniques used to study the various OSNs in mice, as well as in other animal species, such as the nematode (*Caenorhabditis elegans*) or humans (*Homo sapiens*), are diverse and complementary. These include, for example, electrophysiology and imaging techniques at the cellular and tissue levels, as well as physiological and behavioral assays at the animal level [15,16,17,18]. Although these specific methods provide fundamental insights into the functioning of these sensory cells, they require significant resources in research laboratories, such as dedicated and expensive instrumentation, the expertise of the experimenters, and the availability of biological samples (especially for human tissues). Moreover, the inherent physiological characteristics of OSNs can introduce variability in experimental outcomes, which may be influenced by the specific experimental context, potentially leading to distinct interpretations and challenges. For instance, physical access to neurons and sensory tissues can sometimes prevent or compromise the completion of an experiment. For example, electrophysiological recordings of GG neurons are difficult due to their wrapping in protective glial cells [9,14,19,20]. Similarly, in vivo experiments on the VNO require direct physical contact between the animal and the pheromone source, making the standardization of these experiments highly challenging [8,21].

To address these issues, the use of neuronal activity markers, such as the immediate early gene products c-Fos and other members of the family of leucine-zipper transcriptional regulators, seems to be an interesting option [22,23]. Indeed, these markers, following in vivo stimulation with odorants selected by the experimenter, coupled with histology and immunohistochemistry techniques, would allow the precise localization of activated neurons. Thus, tracing and mapping various OSNs across the different olfactory subsystems would become accessible through microscopic analysis. However, this approach is limited by certain constraints. Indeed, aside from the olfactory bulb, the primary cerebral relay for olfactory integration, these activity markers have so far only been reported as effective at the OSN level when used as RNA probes in in situ experiments [21,24,25,26], thus limiting the information on neuronal activity to the transcript level as well as to further protein co-labeling. In addition, in vivo exposure to odorants is difficult to normalize due to the variability in the animal’s innate behavior and movement (neutral cues vs. attraction or repulsion cues), which can result in inconsistent exposure across different tests. Moreover, some olfactory stimuli, especially those associated with warning signals (alarm pheromones, kairomones, smell of smoke, etc.), may induce stress in the animal, further complicating experimentation, and would reasonably raise ethical concerns [27].

Based on these observations, we developed here an alternative and complementary biological assay to existing functional investigation techniques, centered on the phosphorylation of ribosomal protein S6 (rpS6) [28], recently described as an effective marker of activity in OSNs following sensory stimulation [29,30]. We were able to visualize and precisely analyze neuronal-induced stimulations by specific odorants in the GG and highlight the impact of conditioning factors, such as temperature, on its sensory sensitivity. We also extended our observations to the mouse MOE and VNO, as well as to human OSNs, confirming that the rpS6 signal could serve as a robust and reproducible marker of olfactory neuronal activity. We discuss the versatility of the assay for potential applications in other olfactory systems, ligand identification and screening, environmental toxicity assessments, and as a human diagnostics tool, along with its inherent limitations.

## 2. Results

### 2.1. Conceptualization of the Ex Vivo Assay and Validation of the rpS6 Signal as an Indicator of Neuronal Activity in the GG

To develop our new assay, based on the phosphorylation state of ribosomal protein S6, we chose to focus first on the OSNs of the GG as they have their exposure to chemical stimuli regulated by their morphological context, and their sensory sensitivity is dependent on temperature variations [9,12,14,31]. Thus, this dual neuronal response to both chemical and physical stimuli render them particularly interesting to challenge the sensitivity of our assay. Conceptually, we also considered the possibility of extending its application to other olfactory systems.

Our general experimental design includes the following phases (Figure 1a):After sacrifice, the animal’s head is carefully dissected (removing the lower jaw, palate, and skin) and placed in an artificial cerebrospinal fluid (ACSF) solution and conditioned at different temperature settings for 45 to 60 min, a duration necessary to establish cellular equilibrium (transcriptional, translational, and biochemical) [32,33]. A slight hydraulic vacuum is applied along with manual agitation, followed by visual inspection to ensure proper immersion of the head and nasal cavity.Maintaining these conditions, a chemical stimulation is then performed in solution for 45 to 60 min, the time required for rpS6 post-translational modification [29].While keeping these same conditions, chemical fixation in a 4% paraformaldehyde solution (PFA 4%) is applied for 45 to 60 min.A 24-h post-fixation phase in PFA 4% is then performed at 4 °C to further fix the biological tissues.The different olfactory subsystems are then processed individually based on their specific requirements (e.g., microdissection and decalcification treatments).Semi-thin histological sections of the olfactory organs are made.Immunohistochemistry on floating sections is performed to detect the rpS6 signal.Confocal microscopy acquisitions are performed under calibrated settings specific to each conditioning treatment, and representative images are subsequently processed following a standardized protocol.A rationalized qualitative and quantitative analysis is finally conducted.

We used a mouse line expressing green fluorescent protein (GFP) under the control of the olfactory marker protein (OMP) promoter [34] to establish the feasibility of our assay. These OMP-GFP mice allow us to precisely localize all mature OSNs across the different olfactory subsystems. In the first series of tests, where we set the temperature to 23 °C (step 1 of the assay), at which GG neurons are naturally active [13,14,31], and without odorant stimulation (step 2 of the assay), we determined the optimal dilution of the primary anti-rpS6 antibody (1:5000; Figure 1b). This dilution provided a clear rpS6 signal without saturation and with minimized background noise compared to a negative control (without anti-rpS6; Figure 1c). To verify that this signal, linked to post-translational modification of ribosomal protein S6, was compatible with potential protein co-labeling, we then repeated this immunohistochemistry procedure on the GG of C57BL/6 wild-type mice (BL/6), this time incorporating anti-OMP antibodies to highlight OMP expression (Figure 1d). As expected, we observed co-labeling, demonstrating that our assay is suitable for multiple protein visualization.

We then decided to evaluate whether the detected rpS6 signal was sufficiently sensitive to distinguish differences in neuronal activity following controlled stimulation. To do this, we took advantage of the sensitivity of GG neurons to temperature variations. We repeated our assay using calibrated conditioning temperatures (steps 1 and 2 of the assay at 4 °C, 23 °C, 30 °C, or 37 °C; Figure 1e) on both OMP-GFP and BL/6 mice. Visually, we observed that the rpS6 signal was temperature-dependent, with the highest intensity observed at 4 °C. Based on this, we calibrated the acquisition settings for the 4 °C condition and applied these settings across the other temperature conditions (step 8 of the assay). GG neurons exposed to 4 °C displayed a significantly stronger rpS6 signal intensity, while at 37 °C, the signal was markedly weaker (Figure 1f). This not only confirmed previous findings obtained via alternative methods [13,14,31] but also highlighted the sensitivity of our assay.

Remarkably, this sensitivity was also fine enough to reveal differences based on the genetic background of the animal. By conducting a more in-depth analysis of our data, we observed that in knock-in OMP-GFP mice, where *Omp* was genetically replaced by *Gfp*, the rpS6 signal did not show any significant difference between 4 °C and 23 °C (*U*-test; *p* = 0.9238, ns; Appendix A). This observation confirms that the absence of the OMP protein affects the sensitivity and sensory discrimination of OSNs [29,35].

Based on these initial observations, we can thus validate the rpS6 signal as an indicator of neuronal activity in the OSNs of the GG. However, its activation by odorant stimulation remained to be demonstrated.

### 2.2. Establishment of the rpS6 Signal as a Marker of Odorant-Induced Neuronal Activity in the GG

To assess the sensitivity of our assay to odorants, we next exposed GG neurons with two of its recognized chemical ligands at different conditioning temperatures (Figure 2). We focused on 2,4,5-trimethylthiazoline (TMT), a kairomone found in fox (*Vulpes vulpes*) feces, and 2-*sec*-butyl-4,5-dihydrothiazole (SBT), an alarm pheromone emitted by mice in distress [10]. These two molecules have the advantage of also being detected by specific OSNs in the MOE and VNO, respectively [21,36,37,38].

After conditioning OMP-GFP mouse olfactory tissues (step 1 of the assay) at various temperatures, we then applied an odorant stimulation (step 2 of the assay) using either ACSF (Ctrl; Figure 2a) as a non-stimulated reference control [9], or TMT (1:1000; Figure 2a) or SBT (1:1000; Figure 2a) as sources of olfactory stimulation. To precisely observe the effect of odorant stimulation on the rpS6 signal and to eliminate the confounding factor of temperature, we calibrated the acquisition settings based on the odorant-stimulated conditions for each temperature set (step 8 of the assay). This approach allowed us to specifically demonstrate a significant increase in the rpS6 signal in response to both olfactory stimuli (TMT and SBT), independent of the conditioning temperature (Figure 2b). Consistent with our previous observation (Figure 1), we noticed that as the conditioning temperature increased, the rpS6 signal intensity in the GG decreased. The calibration of acquisitions performed at higher temperatures (30 °C and 37 °C), essential for isolating the variations in the rpS6 signal solely linked to odorant stimulation, is therefore based on a weaker rpS6 signal. This directly results in an increase in the overall background noise of the signal (Figure 2a) and may also potentially reduce the consistency of the observed signal across temperatures (Figure 2b). Furthermore, it is important to note that the olfactory sensitivity of the GG naturally diminishes at higher physiological temperatures [39], a characteristic influence that could also amplify this side phenomenon.

These observations not only confirm that our assay can reliably detect GG neuronal activity directly related to olfactory stimulation, but also account for potential experimental and environmental variations, such as proximity to the odorant source or temperature fluctuations, that the animal may encounter in vivo [21,27].

### 2.3. Application of the rpS6-Based Ex Vivo Assay to Other Olfactory Subsystems in Mice

We verified that this assay could be applied to other olfactory subsystems, such as the MOE and the VNO, of BL/6 and OMP-GFP mice (Figure 3). TMT and SBT are known to be detected by specific OSNs in the MOE and VNO, respectively [36,37,38]. While the accessibility of these solubilized molecules to the GG seems expected due to its anterior location, their diffusion to the OSNs of the MOE and VNO appears less evident in the absence of active sniffing or VNO pumping. To assess the apparent accessibility of odorant molecules to the various olfactory subsystems in our assay, we performed a fluorescence test based on the use of the Lucifer yellow dye [9,40] (Appendix A). To simulate odorant accessibility, we incubated mouse OMP-GFP head preparations, along with their tails (used as an impermeable tissue control), in an ACSF solution containing Lucifer yellow at 23 °C during the conditioning and stimulation phases (steps 1 and 2 of the assay). After processing the various tissues, we observed that permeability to Lucifer yellow was clearly detectable in the GG (Appendix A), confirming that hydrosoluble molecules can indeed access the OSNs in this structure [9]. As a control, we confirmed that this hydrosoluble molecule did not penetrate the skin of the mouse tail (Appendix A), as the keratinized epithelium of this tissue is naturally impermeable. Additionally, we observed that Lucifer yellow could reach the sensory epithelia of the MOE (Appendix A) and the VNO (Appendix A), indicating that hydrosoluble odorant molecules access these subsystems in our rpS6-based ex vivo assay. However, we did not conduct an exhaustive inspection of the entire nasal cavity. Unlike other methods, such as direct perfusion on sections of sensory tissues, our assay may not ensure complete access to fluidic stimuli. This limitation should be considered based on the type of odorant molecules investigated and the location of the sensory regions of interest, such as the most posterior regions of the MOE and the VNO. A significant improvement to our assay could be achieved by extracting the various olfactory subsystems from the nasal cavity and meticulously dissecting them before proceeding with conditioning (step 1 of the assay). The generation of tissue sections could also be considered. However, this enhanced accessibility should be weighed against the additional time required for this procedure and the potential disruption of the endogenous physiological configuration, which may be partially altered.

Given the morphological differences between these olfactory organs, after conditioning, stimulation, and fixation (steps 1–4 of the assay), we carried out specific steps of microdissection and decalcification treatment (step 5 of the assay) [29]. We then adjusted the calibration settings for each organ of interest during the acquisitions (step 8 of the assay). During the acquisition of our data, we were surprised to find that the rpS6 signal was not only relevant but also did not depend on the conditioning temperature, unlike the OSNs of the GG. However, the quality of the tissue was unfavorably influenced by an increase of temperature. Indeed, the slices of the MOE (Figure 3a and Appendix A) and the VNO (Figure 3c and Appendix A) obtained at 4 °C and 23 °C exhibited the best morphological preservation, while at higher temperatures (>30 °C), the tissue structure was compromised, making it difficult to prepare histological sections. This is an indirect observation that is shared by other ex vivo assays [10,16,33]. This aspect could likely be optimized by adopting alternative approaches, such as the direct removal of the vomer bone or the VNO capsule [41]. For the MOE, improvements could include the use of younger mice (<6 days old), detachment of the nasal septum from the epithelium [42], or the use of thin cryostat sections, thereby eliminating the need for decalcification required for our floating sections [29].

Nonetheless, in the MOE stimulated by TMT at 4 °C and 23 °C (TMT during step 2 of the assay; Figure 3a and Appendix A), we visually observed a substantial stimulus-dependent activation of OSNs, characterized by a stochastic yet reproducible pattern across experiments. Compared to the non-stimulated control (ACSF during step 2 of the assay; Figure 3a and Appendix A), where endogenous activity was also observed [29], we noted that both the number of OSNs and the rpS6 signal were significantly increased in response to TMT stimulation (Figure 3b and Appendix A), irrespective of the temperature conditioning applied (Appendix A). These observations confirm the specificity of this odorant for certain olfactory receptors (ORs) expressed on targeted OSNs [37,43]. Similarly, in the VNO stimulated by SBT (SBT during step 2 of the assay; Figure 3c and Appendix A), a significant increase in activity was observed compared to the non-stimulated control (ACSF during step 2 of the assay; Figure 3d and Appendix A) regardless of the conditioning temperature (Appendix A). This increase was particularly pronounced in the apical region of the epithelium (Figure 3c), potentially corresponding to the vomeronasal type-1 receptor (V1Rs)-expressing OSNs [38], thus indicating that the OSN targets of SBT in this olfactory subsystem could possibly be identified using the rpS6 marker. Additionally, we found that certain GFP-negative cells in the MOE and VNO, such as non-sensory or basal cells [44], exhibited an rpS6 signal (asterisks in Figure 3c and Appendix A). This suggests that our rpS6-based assay could also be used to investigate the activities of non-neuronal and immature cell populations.

Taken together, these observations demonstrate that our ex vivo assay can be considered to explore sensory activities of various types of OSNs across different olfactory subsystems in mice.

### 2.4. Application of the rpS6-Based Assay to the Human Olfactory System

Finally, we used our assay on biopsies from human olfactory epithelium, as this approach is crucial for advancing current research [15,16,45] and offers the potential to overcome some practical constraints, such as the timely transfer of material between clinical and research settings [45]. We collected biopsies from a human specimen donated through a body donation program. The samples were taken from the posterosuperior aspect of the nasal cavity, near the emergence of the nasal branches of the anterior ethmoid nerve through the cribriform plate. We considered that the OSNs from the olfactory epithelium have been naturally stimulated by ambient odorants perimortem and that, even without undergoing the conditioning and olfactory stimulation steps of our assay (steps 1 to 3), activated OSNs would still be present [29], as we have observed for our MOE and VNO investigations in mice (Figure 3). Therefore, we proceeded with the subsequent steps of the procedure (steps 5 to 9) on these biopsies and initially verified the absence of endogenous immunoreactivity of our antibodies by performing a negative control (without primary antibodies; Figure 4a). We then assessed the cellular identity of the olfactory epithelium by performing targeted labeling against OMP (Figure 4b) and cytokeratin 18 (CK18; Figure 4c), which are respective markers for OSNs and sustentacular cells [46,47]. We observed that, although the signal of our anti-OMP antibody was distinct from the negative control (Figure 4a), its specificity appeared to extend to other cell types beyond OSNs, including non-neuronal cells as well as cells underlying the cartilage and those of the basal lamina (Figure 4b), in contrast to what was observed in mouse olfactory tissues (Figure 1c,d). Thus, the use of other anti-OMP antibodies or alternative neuronal markers, such as the protein gene product 9.5 (PGP9.5) or the beta tubulin III (Tuj-1), could be considered for colocalization studies on the human olfactory epithelium (OE) [42]. On the other hand, staining sustentacular cells with the CK18 antibody proved to be remarkably reliable, validating not only the olfactory identity of our human tissue biopsy but also its state of cellular preservation. Finally, we performed a labeling against rpS6 (Figure 4d) and observed the presence of rpS6-positive cells distributed both in the sensory epithelium as well as in the proliferating basal region [44] (yellow and white asterisks, respectively, in Figure 4d), confirming the cellular endogenous activation of these cells and demonstrating that our assay can be applied to the human olfactory system.

Taken together, we have here demonstrated that our assay, based on the observation of the rpS6 signal under ex vivo conditions, allows for the measurement of neuronal activity in different olfactory subsystems in mice and could also be applied to the human olfactory system. This approach provides an opportunity to explore not only the neuronal activity of different OSNs but also to overcome certain constraints and limitations that in vivo stimulation may pose.

## 3. Discussion

Olfactory impairments are challenging to conceptualize but represent debilitating conditions that significantly impact the quality of life of affected individuals and are often associated with depressive symptoms [48,49]. Following the recent global COVID-19 pandemic caused by the SARS-CoV-2 coronavirus, not only has the scientific community become more aware of the importance of this sense, but so has the general public and authorities [50,51]. Anosmia and other olfactory disfunctions, as well as the role of the olfactory system as a viral entry point, for example, have emerged as crucial concepts, highlighting the need to better understand this system in order to develop effective protective diagnostics and therapeutic measures [46,48,52,53,54,55,56,57]. Studying olfactory neurons in their natural state thus represents a significant scientific challenge, but also a medical necessity, whether the olfactory tissues come from biopsies of living animals or recently deceased human specimens [45,58,59]. Our rpS6-based assay, with its possible experimental modulations, offers a versatile platform to address numerous scientific questions while remaining accessible to fundamental and medical research laboratories. It offers a wide range of potential applications and advantages, which could serve as an alternative or complementary approach to current functional study methods.

The analysis of known odorants, as well as the identification of new chemical structures, such as the pheromones or kairomones involved in olfactory communication [60], could be envisaged with our approach. Whether it concerns specific olfactory subsystems in mice or those from other animal species, including humans, our ex vivo assay would allow the acquisition of information on OSN activity while coupling these data with other indicators. One of the strengths of our method lies in the ability to precisely locate, at the cellular and protein level, the OSNs activated by specific stimuli. Through techniques such as co-labeling (protein–protein or protein–RNA), this rpS6-based approach would allow for the identification of the molecular characteristics of these neurons, including the types of receptors they express, as well as their enzymes or channels, which is particularly relevant for deorphanization of new OR–odorant pairs [61,62,63,64,65,66].

From a methodological standpoint, it is important to note that our assay adheres to the 3Rs principles (reduce, replace, refine) [27,67]. Although the use of animals is still necessary, our approach significantly reduces the number of animals required for experimentation. The ability to precisely condition the olfactory tissue prior to stimulation and refine the process decreases the variations observed between trials. Additionally, this method eliminates the stress experienced by the animal directly related to in vivo stimulations, by replacing it with consistent ex vivo stimulations, while preserving the original structure and morphology of the olfactory subsystem. This also helps to minimize potential variations related to the animal’s innate behaviors toward an odorant molecule, such as avoidance, attraction, or disinterest, which could hinder the standardization of collected results.

In our assay, we tested the influence of temperature variations and demonstrated the impact of this environmental factor on the sensitivity of odorant molecule detection at the Grueneberg ganglion level. Indeed, GG neurons are naturally sensitive to such environmental variables [14,31], which influence their chemodetection [13,18,68]. This physiological aspect is shared by other sensory cells and observed in other species [69,70,71,72,73,74]. In the olfactory system, other OSNs or their cerebral integration have also been reported to be sensitive to various physico-chemical factors, such as pressure variations [75,76], the surrounding gas composition [77,78,79,80], or the velocity and viscosity of the air carrying olfactory information [81,82]. By extension, our assay could be adapted to these parameters by using specific conditioning, thus facilitating these investigations while limiting intrinsic variations.

Finally, it is worth noting that the preservation of living tissues is a key factor in obtaining robust and reproducible data. In our assay, particularly for the mouse MOE and VNO, we observed that low conditioning and stimulation temperatures resulted in better histological sections, while effectively conserving neuronal activity. This observation is particularly interesting as it offers compatibility with biological assays requiring lengthy preparations or prolonged execution times. This could be especially advantageous in cases where there is a significant delay between tissue collection and the biological assay, such as during biopsies of live animals from external sites to research laboratories, or in the context of horizontal transfers of human biopsies between clinical and fundamental research teams, which must be conducted according to strict standards, including the transportation of material on ice [83,84]. From this perspective, it is also important to highlight that, beyond the specific study of OSN activity, our assay could be extended to investigate the impact of potential neurotoxic agents on olfactory neurons [85], as well as the effects of degenerative diseases related to direct or indirect olfactory dysfunction, such as Alzheimer’s or Parkinson’s diseases [86,87]. Furthermore, our ex vivo approach could provide a complementary method for assessing olfactory function in addition to olfactory testing and fMRI studies, specifically for investigating the mechanisms underlying hyperosmia, hyposmia, and parosmia at the level of the olfactory mucosa [88], or even to the tropism of certain pathogens, such as SARS-CoV-2 [45], as our method allows for the artificial maintenance of the sensory epithelium, including OSNs and sustentacular cells, which are viral entry points for this virus [45,46,55,89,90,91,92].

In conclusion, our assay based on the detection of the rpS6 signal, reflecting a post-translational ribosomal modification, constitutes a powerful tool for investigating neuronal activity within different olfactory subsystems, not only in mice but also in other species, including humans [93]. The flexibility of this assay lies in the ability to easily adjust the type of conditioning and stimulation to suit various scientific questions. Furthermore, its compatibility with diverse cellular and tissue biology techniques enhances its potential for application, whether in fundamental research or biomedical studies, thereby opening new perspectives in the exploration of medical diagnostics, olfactory regulation, and communication.

## 4. Materials and Methods

### 4.1. Mice and Human Olfactory Tissus

In this study, both male and female C57BL/6 (*Mus musculus*; Janvier Labs, Saint-Berthevin, France) and homozygous OMP-GFP mice, bred in-house, were used, from 4 to 9 months. In OMP-GFP mice, GFP is expressed under the control of the OMP promoter [34,94], enabling precise localization of OSNs across all olfactory subsystems [95]. Mice were housed under a 12-h light/dark cycle at 21–23 °C in the animal facility and euthanized by cervical dislocation. All experimental procedures complied with Swiss legislation and were approved by the EXPANIM committee of the Lemanique Animal Facility Network and the veterinary authority of the Canton of Vaud (SCAV).

The human specimen was obtained from the body donation program of the Faculty unit of morphology and anatomy, Faculty of biology and medicine, University of Lausanne. The human specimen material was used in accordance with the Guidelines of the Swiss Academy of Medical Sciences.

### 4.2. Sample Preparation, Physiological Solutions, and Chemical Stimulation

After sacrificing the mouse, the head was quickly removed and subjected to an initial dissection in sterile phosphate-buffered saline (PBS; 138 mM NaCl, 2.7 mM KCl, 1.76 mM KH_2_PO_4_, and 10 mM Na_2_HPO_4_; pH 7.4) at room temperature. The lower jaw, the palate, and the skin were carefully removed to allow the entire preparation to be immersed in a Falcon^®^ assay tube (50 mL) containing 35 mL of oxygenated artificial cerebrospinal fluid (ACSF; 118 mM NaCl, 25 mM NaHCO_3_, 10 mM D-glucose, 2 mM KCl, 2 mM MgCl_2_, 1.2 mM NaH_2_PO_4_, and 2 mM CaCl_2_; pH 7.4), freshly prepared at the desired conditioning temperature. A short hydraulic vacuum was applied, followed by visual inspection to ensure the proper immersion of the head and nasal cavity. Temperature regulation was achieved using a water bath with adjustable settings and periodic manual agitation. Olfactory stimulations were performed in a new assay tube of ACSF implemented with TMT (219185; Sigma-Aldrich, Aubonne, Switzerland) or SBT synthesized in-house [10] at a final dilution of 1:1000. Various fixation phases were carried out using 4% paraformaldehyde–PBS solution (PFA 4%; pH 7.4). For the mouse MOE and VNO, a decalcification solution (PBS–EDTA 0.5 M; pH 8.0) was used for 2–3 days post-fixation [29].

The human tissues were initially preserved via systemic perfusion with a fixative solution (2.9% phenol, 2.1% formaldehyde, 5.0% glycerol, and 22% ethanol). Biopsies of the olfactory epithelium were collected from the septal side, near the emergence of the nasal branches of the anterior ethmoid nerve at the cribriform plate.

### 4.3. Histological Procedure

The various olfactory epithelia from mouse and human samples were embedded in 4% low melting agarose (Sigma-Aldrich; A7002). Coronal sections (80 to 120 μm) were generated using a vibroslicer (VT1200S; Leica, Muttenz, Switzerland) and collected in ice-cold PBS [3]. The olfactory tissue slices were then selected under a fluorescent stereomicroscope (M165 FC; Leica) based on their morphology and endogenous GFP expression, before being stored at 4 °C until use.

### 4.4. Immunohistochemistry and rpS6-Based Signal Analysis

Immunohistochemical staining was performed on free-floating sections [29,46,47]. For single labeling, the tissue slices were blocked at room temperature in a PBS-permeabilization solution containing 10% normal goat serum (NGS; Jackson ImmunoResearch, Cambridge, UK; 005-000-121) and 2% non-ionic detergent (Triton^®^ X-100; Fluka, Aubonne, Switzerland), followed by washing steps with a 5% NGS PBS solution. For co-labeling, a similar procedure was followed using 5% normal donkey serum (NDS; Jackson ImmunoResearch; 017-000-121) and washing steps in a 2% NDS PBS solution. The following primary antibodies were used: rabbit anti-rpS6 (α-rpS6; Cell Signaling, Danvers, MA, USA; 5364; 1:5000), rabbit anti-CK18 (α-CK18; PA5-14263; Invitrogen|Thermo Fisher, MA, USA; 1:50), and goat anti-OMP (α-OMP; Wako; 544-10001, Richmond, VA, USA; 1:800). For signal detection, secondary antibodies were selected based on serum specificity: goat anti-rabbit (goat Cy3-conjugated α-rabbit; Jackson ImmunoResearch; 111-165-144; 1:200), donkey anti-goat (donkey FITC-conjugated α-goat; Jackson ImmunoResearch; 705-095-147; 1:200), and donkey anti-rabbit (donkey Cy3-conjugated α-rabbit; Jackson ImmunoResearch; 711-165-152; 1:200). Control experiments were performed without primary antibodies. For general morphological integrity, nuclei were counterstained with Vectashield^®^ antifade mounting medium with 4′,6-diamidino-2-phenylindole (DAPI; H-1200; Vector Labs, Servion, Switzerland). Acquisition was performed using confocal microscopy (Stellaris 8; Leica), under an objective of 40× and with laser intensities calibrated to standardize signal observation and comparison across conditions. Image reconstruction was carried out using cropped maximum 3D-orthogonal projections (IMARIS v6.3 software; Bitplane) to ensure standardized acquisition thickness for all conditions. The rpS6 signal was analyzed post-acquisition using the open-source NIH ImageJ 1.53a software [96]. The mean intensity of the rpS6-related signal for each GFP-positive OSN was measured individually. A minimum of three sections per animal, from at least two mice per condition, were analyzed. Depending on the experimental series, non-stimulation conditions can be used as reference controls for multiple stimulated conditions. The background intensity was calculated by averaging the signal from representative non-OSN regions and was systematically subtracted from each rpS6 signal intensity measurement. Negative values (rpS6 signal less important than the general background) were considered as an rpS6 intensity equivalent to zero for further analysis.

### 4.5. Odorant Accessibility Assay

The accessibility of odorant molecules was assessed through an immersion test using Lucifer yellow fluorescent dye [9,40]. To do this, mouse heads were dissected according to the procedure used in our assay and placed in a 1 mM solution of Lucifer yellow (Sigma-Aldrich; L0259) diluted in ACSF, during the conditioning and stimulation phases of our assay. Cryosections of 30 μm (GG, VNO, tail) to 100 μm (MOE) were then made from the different organs of interest, and the sections were chemically post-fixed in 4% PFA for 20 min. The sections were then rinsed in PBS and counterstained with DAPI for confocal acquisition and observations.

### 4.6. Statistical Analysis

GraphPad Prism 9.3.1 was used for statistical analysis and dot-plot generation. Data are presented as mean ± standard error of the mean (SEM). The normality and homoscedasticity of the data were assessed using the Shapiro–Wilk and Fisher’s tests, respectively. Based on these results, either a two-tailed unpaired Welch’s *t*-test or Mann–Whitney *U*-test was applied for condition comparisons. Statistical significance was set at *p* < 0.05. Significance levels are indicated as follows: * *p* < 0.05, ** *p* < 0.01, *** *p* < 0.001, ns for non-significant.

## Figures and Tables

**Figure 1 ijms-25-13173-f001:**
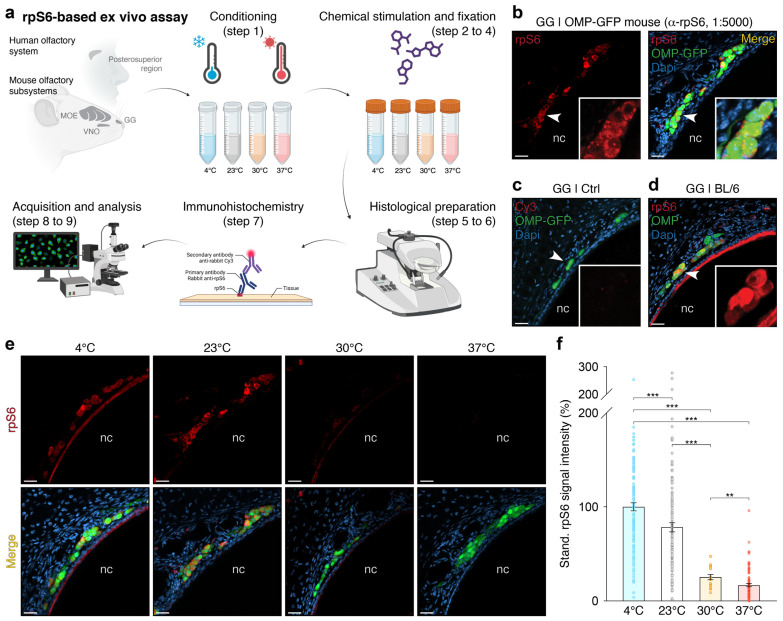
Development of an ex vivo assay based on the rpS6 signal for the assessment of neuronal activity in olfactory systems. (**a**) Schematic representation of the various steps involved in the ex vivo assay. Illustrations were created using BioRender.com (accessed first on 30 November 2022). (**b**) Left panel: representative immunostaining showing the rpS6 signal obtained with the anti-rpS6 antibody (α-rpS6, 1:5000, in red) from the Grueneberg ganglion (GG) of an OMP-GFP mouse, where GG neurons (in green) are visualized based on their endogenous GFP expression. Right panel: the merged view with a nuclear Dapi counterstain (in blue) is shown. (**c**) Negative control performed without the α-rpS6 (Cy3 signal in red). (**d**) Merged view of a double immunohistochemical analysis for rpS6 (α-rpS6, in red) and the OMP protein (α-OMP, in green) in a BL/6 mouse. (**e**,**f**) Assessment of the temperature effect on the rpS6 signal in the GG. Representative immunohistochemical investigations, performed on OMP-GFP mice, are illustrated in (**e**) at different temperatures (4 °C, 23 °C, 30 °C, and 37 °C), and the statistical analysis is shown in (**f**) for both OMP-GFP and BL/6 mice. Nasal cavities are indicated (nc; **b**–**e**). White arrowheads indicate the zoom-in regions (**b**) or the rpS6-related signal zoom-in regions (**c**,**d**). Scale bars: 20 µm (**b**–**e**). Data are expressed as a standardized percentage of the rpS6 signal intensity and represented as the mean ± SEM with aligned dot plots for a minimum of three GG sections per animal, from at least two mice per condition. Comparisons between conditions were performed using two-tailed Welch’s *t*-tests or Mann–Whitney *U*-tests, ** *p* < 0.01, *** *p* < 0.001.

**Figure 2 ijms-25-13173-f002:**
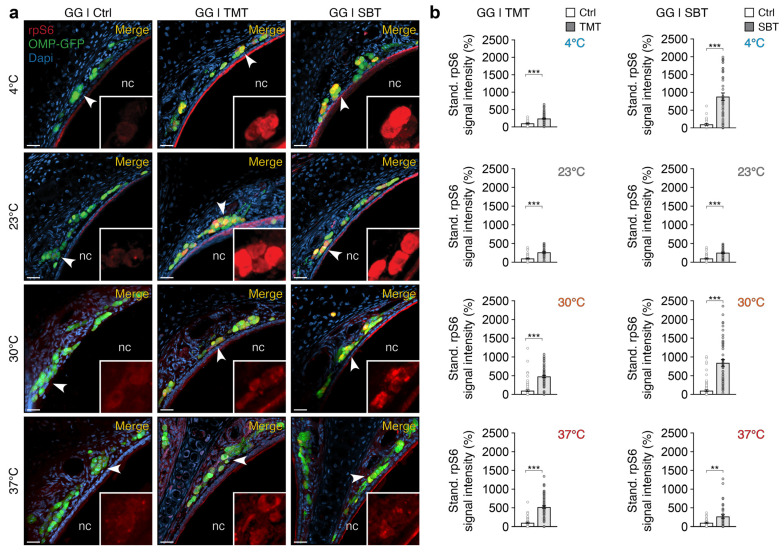
Odorant-induced increases of the rpS6 signal in the GG at different conditioning temperatures. (**a**) Representative immunostaining for the rpS6 signal (in red) in the GG of OMP-GFP mice (OMP-GFP signal, in green) following conditioning at different temperatures (4 °C, 23 °C, 30 °C, and 37 °C) and odorant stimulations, with ACSF serving as the reference control (Ctrl), TMT, or SBT. The merged images include nuclear Dapi staining (in blue). (**b**) The statistical analysis comparing the effects of odorant stimulations calibrated at different conditioning temperatures is displayed. Stimulations with the Ctrl and odorants (TMT in left panels and SBT in right panels) are represented by white and gray bars, respectively. Nasal cavities (nc) are indicated in (**a**). White arrowheads highlight zoomed-in regions of the rpS6 signal (**a**). Scale bars: 20 μm (**a**). Data are presented as a standardized percentage of the rpS6 signal intensity, with the means ± SEM displayed using aligned dot plots. A minimum of three GG sections per animal from at least two mice per condition were analyzed. Comparisons between conditions were performed using two-tailed Mann–Whitney *U*-tests, ** *p* < 0.01, *** *p* < 0.001.

**Figure 3 ijms-25-13173-f003:**
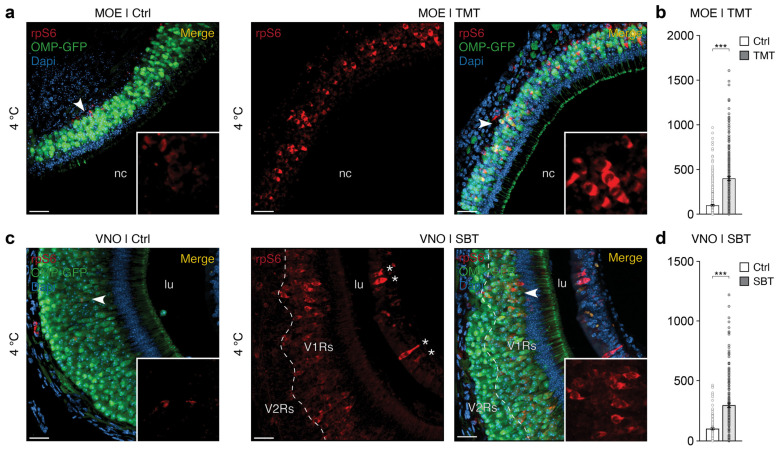
Investigation of rpS6-based odorant signals in the MOE and VNO of mice. (**a**) Here, representative immunostaining for the rpS6 signal (in red) in the MOE of OMP-GFP mice (OMP-GFP signal, in green) under non-stimulated control conditions (Ctrl, left panel) and after TMT stimulation (TMT, right panels) are shown here for a conditioning temperature of 4 °C. (**b**) Statistical analysis comparing the effects of TMT stimulations on GFP-positive OSNs in the MOE. Data were combined across the conditioning temperatures of 4 °C and 23 °C. (**c**) Here, representative immunostaining for the rpS6 signal (in red) in the VNO of OMP-GFP mice under non-stimulated conditions (Ctrl, left panel) and after SBT stimulation (SBT, right panels), are shown for the same conditioning temperature of 4 °C. The approximate boundaries between the vomeronasal type-1 receptor (V1R) and vomeronasal type-2 receptor (V2R) layers are marked with a white dashed line. Non-sensory GFP-negative cells expressing the rpS6 signal are indicated by white asterisks. (**d**) Statistical analysis comparing the effects of SBT stimulations on GFP-positive OSNs in the VNO. Data were combined across the conditioning temperatures of 4 °C and 23 °C. Nasal cavities (nc) in the MOE (**a**) and lumen (lu) in the VNO (**c**) are annotated. White arrowheads highlight regions where the rpS6 signal is zoomed in (**a**,**c**). Scale bars: 20 μm (**a**,**c**). Merged images include nuclear Dapi staining (in blue, (**a**,**c**)). Stimulations with the Ctrl and odorants are represented by white and gray bars, respectively (in (**b**,**d**)). A minimum of four tissue sections per animal from at least three mice per condition were analyzed. Comparisons between conditions were performed using two-tailed Mann–Whitney *U*-tests, *** *p* < 0.001.

**Figure 4 ijms-25-13173-f004:**
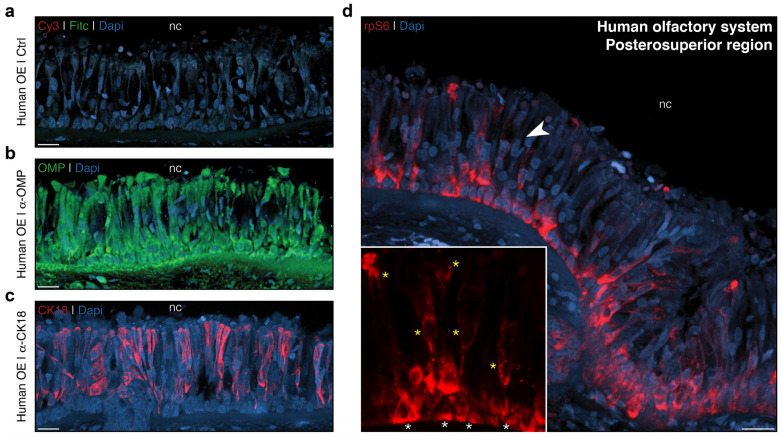
Investigation of rpS6-based signals in the posterosuperior region of the human olfactory system. (**a**) Negative control performed on the posterosuperior region of the human olfactory epithelium (OE) without primary antibodies, illustrating the endogenous signals related to Cy3 (in red) and FITC (in green). (**b**) Representative immunostaining showing the α-OMP antibody signal (in green) in the human OE and its apparent lack of neuronal specificity. (**c**) Representative immunostaining for the α-CK18 antibody signal (in red) performed in the same region, highlighting its specificity for the sustentacular supporting cells of the olfactory epithelium. (**d**) Labeling for the rpS6 signals (in red) in the posterosuperior region of the human OE. Nasal cavities (nc, (**a**–**d**)) are indicated. The white arrowhead highlights a zoomed-in region of the rpS6 signals, within which white and yellow asterisks indicate basal cells and sporadic cells of the sensory epithelium, respectively (**d**). Scale bars: 10 μm (**a**–**d**). Dapi staining is used as a nuclear marker (in blue, (**a**–**d**)).

## Data Availability

The original contributions presented in the study are included in the article; further inquiries can be directed to the corresponding contacts M.-C.B. and J.B.

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
