# Peer review of "Development of an rpS6-Based Ex Vivo Assay for the Analysis of Neuronal Activity in Mouse and Human Olfactory Systems"

_ijms, 2024, doi:10.3390/ijms252313173_

Round 1

Reviewer 1 Report

Comments and Suggestions for Authors

Broillet Olivier et al., present an assay that utilizes the phosphorylation status of the ribosomal protein S6 as an activity marker of olfactory sensory neurons following temperature stimulation and chemical stimulation with SBT and TMT of ex vivo tissue preparations obtained from mouse and human. This assay is supposed to overcome limitations associated with the exposure to olfactory stimuli under in vivo conditions. Therefore, mice were sacrificed, their heads prepared in PBS, and conditioned in oxygenated ACSF at four different temperatures, prior to stimulation with SBT or TMT and immunohistochemical analyses of the GG, the VNO and the MOE. Human olfactory biopsies were obtained post mortem following systemic fixation perfusion.

The manuscript is well written and structured regarding the GG analyses (part 1), but the story falls behind regarding the data obtained in the murine VNO and MOE (part 2), and the human MOE (part 3), which makes it challenging to accept the proposed broad application of this method as it is. Thus, the benefit of this procedure in comparison to current techniques using for example stimulation of ex vivo tissue slice preparations is not yet clear.

This impression is mainly due to the very well-analyzed data set obtained in the GG. The authors confirm earlier results from the Fleischer’s and their own lab, which demonstrated that GGNs show temperature sensitivity with a maximum of Fos activation at cool ambient temperatures and analyses regarding the GG ligands TMT and SBT. Here in the new study, the ex vivo experiments conducted by Broillet Olivier et al., using the activity reporter pS6 yielded similar results, confirming the robustness of the assay for the GG. However, the story changes when considering the new application of pS6 immunoreactivity in preparations of the murine VNO/MOE and the human MOE. The complete lack of any statistically reliable data makes it difficult to accept the proposed general application of this method.

MAJOR CONCERNS

It is essential to provide statistical data on the mouse VNO and MOE

Lines 243 ff.: “… higher temperatures (>30°C), the tissue was compromised …”.  

The authors state that statistical analyses for pS6 immunohistochemistry in the VNO and MOE are hindered by higher temperatures and prolonged decalcification times, which compromise tissue morphology and quality. However, the VNO does not require decalcification. Many groups, including your own lab (doi: 10.3791/3311), routinely remove the vomer bone/capsule when working with acute VNO slice preparations. This allows them to conduct calcium imaging analyses or electrophysiology in thick ex vivo vibratome sections. The same is true for the MOE. Vibratome sectioning works well with young mice (up to 6 days of age). An alternative method is to dissect the nasal septum and peel off the epithelium in older mice. Furthermore, thin MOE cryosections can be prepared from young adult mice without decalcification. It is unclear to me why there are no pS6 statistics on the two murine subsystems in the control and stimulated specimens. Referring again to lines 243 ff., you should include at least statistics on the data sets you have obtained at the lower temperatures (4°C and 23°C) showing reasonable tissue integrity.

Stimulus access

How is the access of the stimulus fluid to the different subsystems provided in your preparation? If I understood right from your method section (lines 115 – 117 and lines 400 ff.), the entire preparation is submerged into stimulus fluid for 45 – 60 min. For the GG, access is likely provided by the most anterior location of this organ, a short distance. However, for the VNO and MOE, access of the stimulus fluid it is not entirely clear to me since neither VNO pumping nor active sniffing is possible. In contrast to the GG, the access of the stimuli to the VNO via the vomeronasal duct and to the MOE situated in the posterodorsal aspect of the nasal cavity is rather long considering time and distance. Please elaborate on this question. Do you see a gradual decline of the pS6 signal regarding the anterior to posterior extent of the VNO / MOE? In line with this question, again statistics on the pS6 signal is essential.

OMP-GFP genotype

Lines 175–180: It is notable that mice homozygous for GFP, which are in fact OMP-knockout mice, do not exhibit differences in temperature sensitivity of the GG between 4°C and 23°C. Please provide a bar histogram of the pS6 signal, maybe included in Fig. 2. This information is valuable for the olfactory community and provides further insight into the function of OMP. Please also indicate in the method section which OMP-GFP mouse genotypes (hetero/homozygous) were used for the different experiments.

Fig. 2: This figure will be complete once the authors have included the pS6 immunohistochemistry for the GG at 30°C and 37°C under control, TMT, and SBT conditions. From my perspective, this is crucial because the pS6 signal intensities quantified in the plots shown in Fig. 2b at higher temperatures are also markedly different from the identical ones at lower temperatures using TMT. In the case of the SBT treatment the situation is even more complex. It displays the lowest pS6 intensities at 23°C and 37°C, and the highest pS6 intensities at 4°C and 30°C. This valuable information needs to be shown. Extending on this, I suggest that the y-axis scaling for all plots shown in Fig. 2b must be identical to facilitate visualization of the effects at the different temperatures.

Human MOE

Fig. 4: The specificity of the OMP staining shown in Fig. 4b is questionable and has not been described thoroughly in either the results or the legend. Fig. 4b lacks the requisite arrows indicating OMP-positive OSNs. Every single cell is labeled, including the basal lamina and the cells of the underlying cartilage. To prove specificity, you must conduct double-labeling IHC of OMP and CK-18. This will prove the specificity of your OMP staining, as it seems that sustentacular cells are also positive for OMP. The lack of OMP specificity in Fig. 4b calls the reliability of the co-localization shown in Fig. 3d into question. Please explain why only cells of the basal MOE layer show pS6 staining, giving the impression of a necklace.

Reviewer 2 Report

Comments and Suggestions for Authors

In this study, the authors successfully demonstrate that phosphoserine 6 (RPS6) can be used as a neuronal marker of mouse Grueneberg ganglion neurons (GGNs) activities in ex-vivo. They found that RPS6 signals in GGNs were temperature-dependent, which is consistent with the function of GGNs as thermosensors. Further, they found that RPS6 signals in GGNs increased in response to odorants, such as a kairomone (TMT) and an alarm pheromone (SBT). Additionally, they report that TMT stimulation increased RPS6 signals in olfactory sensory neurons (OSNs) within the main olfactory bulb (MOB), while SBT stimulation increased RPS6 signals in the vomeronasal organ (VNO). Unlike GGNs, they did not observe temperature-dependent RPs6 signal changes in OSNs within MOB and VON. Finally, they show that their ex-vivo assay can also be applied to biopsies from human olfactory epithelium. I think their assay based on the RPS6 signal detection can be useful in overcoming certain constraints and limitations in in-vivo preparations. I have only minor comments below.

Temperature dependency of RPS6 signals in GGNs is clearly demonstrated in Fig. 1. However, this dependency does not appear in the control for TMT and SBT odorants, especially for SBT in Fig. 2. What makes the temperature dependency unclear?

Similar to TMT, I expect SBT stimulation to show consistent signal changes regardless of temperature. Could the smaller response at 37°C be due to an unfavorable influence from the higher temperature?

Round 2

Reviewer 1 Report

Comments and Suggestions for Authors

Manuscript has been improved significantly.